# Heterosis and Combining Ability Analysis of Fruit Yield, Early Maturity, and Quality in Tomato

**Zengbing Liu, Jingbin Jiang, Ai Ren, Xiangyang Xu, He Zhang, Tingting Zhao, Xiuming Jiang, Yaoguang Sun, Jingfu Li \* and Huanhuan Yang \***

College of Horticulture and Landscape Architecture, Northeast Agricultural University, Harbin 150030, China; liuzengbing2019@163.com (Z.L.); jjb1248@126.com (J.J.); renai1995@163.com (A.R.); xxy709@126.com (X.X.); iamzhe2003@yahoo.com (H.Z.); ttzhao2016@163.com (T.Z.); jxm1100101@163.com (X.J.); 18846816370@163.com (Y.S.)

\* Correspondence: lijf_2005@126.com (J.L.); huanyaya0126@sina.com (H.Y.)

**Abstract:** Heterosis and combining ability are two important considerations in the utilization of heterosis, which can be used to generate excellent hybrid resource candidates and is very important in conventional hybrid breeding. In this study, the combining ability, heritability, and heterosis of eight major agronomic traits were analyzed in 10 tomato parents and 45 crosses between them. As well as TY-301, a recognized and official excellent variety that is currently selling well on the market was used as a control to conduct a control heterosis analysis, with the goal of selecting ideal parents with high combining ability and new hybrids with commodity value, high yield, early maturity, and high quality. The results showed that both additive and nonadditive genetic effects are involved in the expression of the traits and that the additive genetic effect is dominant in trait inheritance. Although general combining ability (GCA) and specific combining ability (SCA) were not correlated, and the strength of heterosis depends on SCA, the sum of the parental GCA values (GCAsum) did predict heterosis for some traits with higher predictive accuracy than did SCA. Compared with heterosis, GCAsum can better predict hybrid performance. Finally, the parent 17,969 was the breeding material with the best comprehensive trait performance, especially in yield. We screened a high-yielding candidate combination 17,927 × 17,969 and a precocious and good taste candidate combination 17,666 × 17,927. This information may play an important role in the selection of superior parents and hybrid combinations based on combining ability and heterosis analysis.

**Keywords:** tomato; general combining ability; specific combining ability; heritability; heterosis

## 1. Introduction

Tomato (*Solanum lycopersicum* L.), a staple vegetable crop with global distribution, exhibits obvious heterosis, especially with respect to early maturity and yield [1]. Heterosis, a common natural phenomenon, is the biological basis of crossbreeding; it depends on genetic differences between the parents [2], is the most profound in $F_1$, and gradually diminishes starting in $F_2$. It is an extremely important means of genetic improvement and has been widely used in field crops (e.g., rice and maize), as well as vegetable crops (e.g., species of Cruciferae and Solanaceae) [3]. For example, in Europe, ≈100% of sugar beet, >90% of rapeseed, and >70% of rye varieties are hybrids, as are >70% of rice varieties in China and >80% of varieties cotton in India [4]. Utilizing heterosis in agricultural production can significantly improve yield [5–7], quality [8–10], and resistance [11–13]. The utilization of crop heterosis is considered a landmark innovation of modern agriculture and has yielded great economic gains [4].

Analyzing heterosis and combining ability are two important considerations in the utilization of heterosis; it is the first step in breeding inbred lines to develop commercial hybrids. Progeny selection is one of the most important stages in plant breeding, but producing excellent progeny depends on the chosen parents. Combining ability is useful

for successfully predicting the genetic capabilities of parental lines and crosses [14,15]. Therefore, the level of combining ability is an important criterion for selecting the parents of hybrids, and it directly affects hybrid quality [16]. For some traits, parents with high GCA effects produce hybrids with low SCA effects. On the other hand, parents with low GCA effects can produce hybrids with high SCA effects. Therefore, when selecting elite parents for crosses, understanding and accounting for the relationship between the GCA of the parents, the SCA of the cross, and the dominant type of combining ability are key to improving breeding efficiency based on combining ability [3]. Some studies indicate that there is no fixed relationship between GCA and SCA [17–20]. Xiang et al. revealed a correlation between parental traits and GCA [21], but parental performance is not correlated with heterosis [22]. Therefore, parental performance per se and GCA are not necessarily reliable predictors of heterosis.

Many studies have shown that parental GCA may be a good predictor of hybrid performance [23–26]. The relative contributions of the GCA and SCA effects to the performance of hybrids depend on traits and crosses; for aflatoxin and grain yield [27], the GCA effect is more important than the SCA effect. However, crosses with higher SCA effects were useful for obtaining high-performing hybrids. Therefore, hybrid performance may depend on the GCA and SCA effects, and different traits may be dominated by either of the combining abilities (GCA or SCA). In maize, heterosis correlates with the genetic distance between the parental inbred lines and increasing genetic distances (variability) between parents increases heterosis [2,28]. Heterosis is also related to the heterogeneity and diversity of the parents. The general combining ability is mainly determined by the additive effect of the parental genes, SCA is controlled by dominance and nonallelic gene interactions, and heterosis is mainly caused by dominance and nonallelic interactions, i.e., heterosis mainly depends on the effect of SCA. Therefore, the strength of the SCA in a given cross plays a decisive role in determining the strength of heterosis on the cross' yield [29] and quality [30]. Zhou et al. also have found that the SCA effect in rice is very important for yield heterosis: the loci *Sd1*, *Ghd7*, *Ghd8*, and *DEP1* show a strong GCA effect on many agronomic traits, though not on yield or the seed setting rate; and the QTLs *Ghd8*, *S5* and *qS12* show strong SCA effects on yield and thus overdominance, which also indicates that different traits may rely on different types of combining ability [31]. Since there is no direct relationship between the additive effect of genes and dominance or epistasis, high-yield combinations may be produced in combinations with high GCA or high SCA. Breeding practice shows that the GCA levels of potential crossing parents should be determined beforehand to avoid blind selection, and the selected parents should have high levels of SCA so that high-quality crosses can be obtained.

Although ubiquitous, heterosis does not necessarily occur in every hybridization between two parents or materials, and good crosses do not necessarily come from good parents. Given this variance, the degrees of heterosis in different crosses or different traits within the same cross also vary. Therefore, selecting parents based on past performance does not always produce the desired outcome [32]; the parents need to be judged based on their potential to produce excellent hybrids rather than on their own performance [33]. To identify parents with high crossing potential, their combining abilities should be assessed, and ideal parents and crosses can be selected on this basis. Assessing the strength of heterosis requires determining the portion of the genetic effects of heterosis that are utilizable but not easily fixed between different crosses or traits while evaluating their practical value in heterosis breeding, and this assessment provides a basis for parental choice [16]. Thus, we chose 10 highly inbred homozygous tomato parents with diverse traits as the test material and made 45 crosses by adopting the Griffing IV complete diallel design (P(P − 1)/2), and in order to guarantee the commodity value of the selected combination, we used TY-301 as the control. We analyzed the combining ability, heritability, and heterosis of eight traits, i.e., plant height, early yield, total yield, fruit weight, fruit number per plant, first ripening stage, fruit hardness, and soluble solid content, of the parents and their hybrid progeny. Through this experimental approach, we aimed to screen for crosses with high levels of

heterosis and usability, as well as parental materials with good overall performance, thus providing high-quality germplasm resources for studies on the mechanism of heterosis and to promote the breeding of tomato hybrids.

## 2. Materials and Methods

### 2.1. Materials

Ten highly inbred homozygous parents with indeterminate growth and diverse traits (Table 1), 45 $F_1$ hybrids, and TY-301 were used as the experimental materials. The parent material had either a few large fruits or many small fruits. The hybrids were obtained by crossing using the Griffing IV complete diallel design, $P \times (P - 1)/2$. The control was TY-301, which is a recognized and official excellent commercially available cultivated hybrid with uniform large pink fruits, green shoulders, good hardness, high yield, and medium maturity, among other traits. All materials were provided by the Tomato Research Institute of the College of Horticulture and Landscape Architecture, Northeast Agricultural University, China.

**Table 1.** Traits of test parents.

| Code | Generations | FC | FS [a] | GS | fs [b] | VN | SSC | FH | FFN | *RIN* |
|------|-------------|-----|--------|-----|--------|-----|-----|-----|-----|-------|
| 17,648 | Six | Yellowish | Round/nearly round | Absent | Oversized | 6.5 | 4.7 | 49.84 | 7 | Present |
| 17,666 | Six | Pink | Oblate | Present | Large | 8.5 | 4.9 | 21.4 | 4.8 | Absent |
| 17,719 | Six | Pink | Round/nearly round | Absent | Oversized | 5 | 4.7 | 27.78 | 6 | Absent |
| 17,896 | Six | Pink | Oval/cone | Present | Oversized | 3.5 | 5.3 | 24.43 | 6.4 | Absent |
| 17,904 | Six | Pink | Oblate | Absent | Large | 4 | 4.6 | 21.67 | 6.8 | Absent |
| 17,927 | Six | Pink | Round/nearly round | Present | Medium | 4 | 4.6 | 43.53 | 5.2 | Absent |
| 17,942 | Six | Pink | Round/nearly round | Present | Large | 3.5 | 4.0 | 24.69 | 5 | Absent |
| 17,955 | Six | Pink | Oblate | Absent | Medium | 4 | 4.7 | 33.09 | 6.2 | Absent |
| 17,969 | Six | Pink | Oval/cone | Present | Medium | 4.5 | 4.7 | 43.13 | 7.6 | Absent |
| 17,996 | Six | Pink | Oblate | Present | Medium | 5 | 5.7 | 29.49 | 7.2 | Absent |

FC—Fruit color, FS—Fruit shape, GS—Green shoulder, fs—Fruit size, VN—Ventricular number, SSC—Soluble solid content, FH—Fruit hardness, FFN—First flower node. [a] Oblate: The fruit shape index is between 0.6 and 0.8. Round/nearly round: The fruit shape index is between 0.8 and 0.9. Oval/cone: The fruit shape index is between 0.9 and 1. Obround: The fruit shape index is greater than 1. [b] Medium fruit: The fruit weight is less than 180 g. Large fruit: The fruit weight is less than 220 g and greater than 180 g. Oversized fruit: The fruit weight is greater than 220 g.

### 2.2. Field Experiment and Data Collection

Seeds of the 10 parental lines, 45 hybrids, and control (TY-301) were sown at the Horticulture Experimental Station of Northeast Agricultural University in Harbin (125°42′–130°10′ E, 44°04′–46°40′ E) on 24 March 2019. The seedlings were transplanted to Xiangyang Farm on 29 April 2019 and field-planted in plastic greenhouses on 1 June 2019. In the yield plots, there were 12 plants per row per genotype, the plant spacing was 40 cm, and the row spacing was 30 cm. To reduce deviation, a guideline was fixed on the ground at both ends of each row, and the seedlings were planted along the line at the designated density. A randomized complete block design with three replicates was adopted. During the experiment, normal agronomic management measures for tomato production were used based on field management under a shed. Weeding and all cultivation practices were conducted as needed.

Eight traits were investigated. The first ripening stage, an early-maturity-related trait, was recorded. The plant height was measured at the time of harvest, and the early yield after the harvest, representing another early-maturation-related trait, was measured, along with three yield-related traits (total yield, fruit number per plant, and fruit weight) and two quality traits (fruit hardness and soluble solid content). The measurement standards of the eight traits were as follows:

(1) Plant height (PH)//cm: At harvest, a measuring tape was used to measure five consecutive plants, excluding those at the edges of the plot, from the stem base to the base of the fourth inflorescence in the upright growth state of the plant; the measurement was repeated 3 times, and the average of the measurements was taken.

(2) Fruit weight (FW)//g: This is the total fruit yield of the plant divided by the total number of fruits.

(3) Fruit number per plant (FNPP): This is the total number of fruits from the first, second, third, and fourth inflorescences at maturity. The selected fruits were all smooth and complete or had slight mechanical damage. Deformed fruits infected with disease, rotten fruits and fruits with substantial cracking were not included in the statistical analysis.

(4) Early yield (EY)//kg: This is the total yield after harvesting at 4-day intervals prior to the full fruit period, the period when the yield in the plot was greatest.

(5) Total yield (TY)//kg: This is the total yield of the fruits borne on the first, second, third, and fourth inflorescences of all plants in the plot.

(6) First ripening stage (FRS)//d: This is the number of days from sowing to the maturity of the first ripened fruit on a plant.

(7) Fruit hardness (FH)//N: A peel sample of approximately 1 cm$^2$ was removed with a blade at a 120° angle between the shoulder of each fruit. A probe of 1 cm$^2$ was selected, and a hand-held durometer (HANAPI, MODEL GY-4) was used to measure according to the manufacturer's instructions. Five complete fruits were randomly selected from each plot, and the average value was considered the hardness value of the fruit in the plot; this process was repeated 3 times.

(8) Soluble solid content (SSC)//%: At the full fruit period, five fruits were randomly chosen from each plot. Each fruit was cut crosswise, the juice was squeezed out by hand, and the juice was put into a clean, dry container. Approximately the same amount of liquid was taken from five fruits and mixed well. The content of the combined juice was determined using a digital handheld Atago PAL-1 "Pocket" refractometer.

### 2.3. Statistical Analyses

The GCA/(GCA+SCA) ratio was calculated using an equation from Baker [34] and modified by Hung and Holland [35]:

$$2\sigma^2{}_{GCA}/2\sigma^2{}_{GCA} + \sigma^2{}_{SCA} \tag{1}$$

where $2\sigma^2{}_{GCA}$ is the variance in GCA effects derived from the mean square of the GCA, and $\sigma^2{}_{SCA}$ is the variance in SCA effects derived from the mean square of the SCA. Since the total genetic variance among $F_1$ hybrids is equal to twice the GCA component plus the SCA component, the closer this ratio is to unity, the greater the proportion of a specific hybrid's performance can be predicted based on GCA alone [34].

Heterosis was estimated based on two standards, midparent heterosis and comparison heterosis, using the following formulas:

$$\text{Midparent heterosis (MPH): H} = ((F_1 - MP)/MP) \times 100\%$$

$$\text{Control heterosis (CH): H} = ((F_1 - CK)/CK) \times 100\%.$$

In the formulas, $F_1$ is the mean value of the trait in the hybrid, $MP$ is the mean value of the trait in the two parents, and $CK$ is the mean value of the trait in the control.

Data analysis and processing were conducted using Excel. The variance, combining ability, and heritability were analyzed by statistical analysis program of genetic mating design under professional statistics module in the statistical software DPS 7.05 (Data Processing System). The combining ability effect of the test materials was estimated according to the fixed model (model I), and estimate various variance components and related genetic parameters according to the random model (model II) [20]. Correlation and significance analyses and the associated plotting were performed using Origin 19.0 Software (OriginLab company, Northampton, MA, USA).

## 3. Results

### 3.1. Performance, Variation Coefficient, and Correlation Analysis of Different Traits in 45 Hybrids

The intratrait variation in the three traits (EY, FH, and FNPP) was significant (Table 2). The coefficient of variation in parental hardness (FH) was the highest, which is related to including late-maturing varieties among the parents. The average $F_1$ performance for EY, TY, and FNPP was significantly better than that of the parents, but the opposite was true for FRS, indicating that the $F_1$ hybrid tomato plants showed profound heterosis in yield and early maturity. The nonsignificant difference in SSC and the decrease in FH in the hybrids were likely associated with early maturity. For all traits except SSC, the parents showed a higher coefficient of variation than did their $F_1$ hybrids, indicating that the unstable heterosis of these traits is mainly controlled by the instability of the parents and that in the same environment, $F_1$ hybrids are more stable than their parents.

The results of the correlation analysis are shown in Figure 1. The PH had a significantly negative correlation with FNPP and a significantly or highly significantly positive correlation with FW and FRS, indicating that taller plants had heavier FW, a lower fruit setting rate, and later maturity. The correlations between TY, EY, and FNPP were significantly or highly significantly positive, indicating that FNPP is an important factor in high yield and that an earlier yield helps increase the TY. The FW had a highly significantly negative correlation with the FNPP and a significantly positive correlation with FRS, indicating that plants with high FW bear fewer fruits and mature later. The FW and FRS had a significantly positive correlation with FH and a significantly negative correlation with SSC, indicating that fruits with high hardness are heavier, mature later, and have lower SSC.

**Table 2.** The expression and coefficients of variation in 10 parents, 45 $F_1$ hybrids, and control (CK) for eight main agronomic traits.

| Trait | $F_1$ | | | | P | | | | CK | | | |
|---|---|---|---|---|---|---|---|---|---|---|---|---|
| | Mean ± SD | CV | Order | Range | Mean ± SD | CV | Order | Range | Mean ± SD | CV | Order | Range |
| PH | 114.85 ± 9.37 Aa | 0.08 | 7 | 96.67~132.20 | 116.95 ± 14.07 Aa | 0.12 | 7 | 91.6~132.67 | 91.87 ± 5.53 Bb | 0.06 | 6 | 90.8~94 |
| EY | 5.09 ± 1.32 Bb | 0.26 | 1 | 2.22~8.80 | 4.04 ± 1.27 Aa | 0.32 | 3 | 1.97~6.71 | 5.59 ± 2.69 ABab | 0.48 | 1 | 3.23~8.58 |
| TY | 13.59 ± 1.97 Aa | 0.14 | 5 | 9.75~18.27 | 9.68 ± 1.44 Bb | 0.15 | 5 | 7.76~11.6 | 13.57 ± 1.78 Aa | 0.13 | 5 | 11.58~15.02 |
| FNPP | 13.87 ± 2.98 Aab | 0.22 | 3 | 7.40~18.73 | 10.58 ± 3.69 Ab | 0.35 | 2 | 6.2~19.33 | 17.58 ± 3.97 Aa | 0.22 | 3 | 72~109 |
| FW | 202.73 ± 35.46 Aa | 0.17 | 4 | 147.97~268.80 | 197.91 ± 52.95 ABa | 0.27 | 4 | 105.15~278.65 | 159.78 ± 21.56 Bb | 0.13 | 4 | 137.75~180.83 |
| FRS | 135.67 ± 4.79 Cc | 0.04 | 8 | 122.00~142.00 | 141 ± 8.62 Aa | 0.06 | 8 | 130~156 | 138 ± 0.00 Bb | 0 | 8 | 0 |
| FH | 18.43 ± 4.72 Aa | 0.26 | 2 | 10.37~39.40 | 22.61 ± 10.98 Aa | 0.49 | 1 | 11.96~45.86 | 23.69 ± 5.67 Aa | 0.24 | 2 | 18.31~29.61 |
| SSC | 3.39 ± 0.42 Aa | 0.13 | 6 | 2.20~4.20 | 3.4 ± 0.43 Aa | 0.13 | 6 | 2.7~4.0 | 3.27 ± 0.06 Aa | 0.02 | 7 | 3.2~3.3 |

Lowercase letters and uppercase letters indicate significance at 0.05 and 0.01 probabilities, respectively.

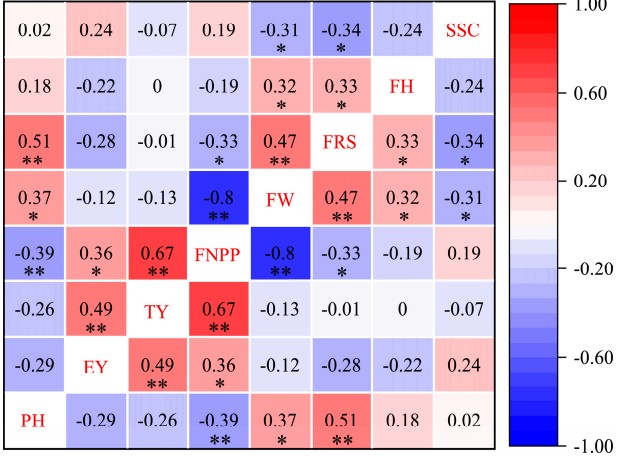

**Figure 1.** A heatmap depicting Pearson's correlation coefficients for different characters of 45 tomato hybrids. Sample size, N = 45. The degree of correlation is color coded according to the color key on the right. The blue and red boxes indicate negative and positive correlation coefficients, respectively. The symbols * and ** indicate significance at 0.05 and 0.01 probability, respectively. PH—plant height, EY—early yield, TY—total yield, FNPP—fruit number per plant, FW—fruit weight, FRS—first ripening stage, FH—fruit hardness, SSC—soluble solid content.

### 3.2. Variance Analysis of Different Traits in 10 Parents and 45 $F_1$ Hybrids

The results of the variance analysis are shown in Table 3. All the traits except EY and TY differed significantly between the parents. In terms of the crosses, there are true genetic differences in the traits between the crosses, whose combining ability could be analyzed using models I and II of the Griffing IV design.

**Table 3.** Variance analysis of eight main agronomic traits in 10 parents and 45 $F_1$ hybrids.

| Trait | Parents | | | | $F_1$ | | | |
|---|---|---|---|---|---|---|---|---|
| | *df* | *MS* | *F* | *P* | *df* | *MS* | *F* | *P* |
| PH | 9 | 593.570 | 15.178 ** | 0.000 | 44 | 263.290 | 6.690 ** | 0.000 |
| EY | 9 | 4.778 | 1.886 | 0.114 | 44 | 5.226 | 1.962 ** | 0.004 |
| TY | 9 | 6.205 | 1.538 | 0.201 | 44 | 11.587 | 4.087 ** | 0.000 |
| FNPP | 9 | 40.747 | 9.830 ** | 0.000 | 44 | 26.661 | 8.603 ** | 0.000 |
| FW | 9 | 8410.921 | 9.455 ** | 0.000 | 44 | 3772.730 | 6.845 ** | 0.000 |
| FRS | 9 | 223.041 | 39.130 ** | 0.000 | 44 | 68.864 | 6.237 ** | 0.000 |
| FH | 9 | 361.569 | 25.008 ** | 0.000 | 44 | 66.705 | 5.525 ** | 0.000 |
| SSC | 9 | 0.556 | 9.063 ** | 0.000 | 44 | 0.540 | 2.810 ** | 0.000 |

Notes: *df*—degrees of freedom, *MS*—mean square, *F*—statistical magnitude, *P*—probability. The symbol ** indicates significance 0.01 probabilities, respectively.

### 3.3. Variance Analyses of Heritability and Combining Ability for Different Traits

A variance analysis of combining ability is a prerequisite that determines whether combining ability can be analyzed. The variances in GCA and SCA for all traits were statistically significant or highly statistically significant, and the $(2\sigma^2_{GCA}/2\sigma^2_{GCA} + \sigma^2_{SCA})$ ratio was close to 1 (Table 4). This result indicates that both additive and nonadditive genetic effects are involved in the expression of the traits, and the additive genetic effect plays a dominant role.

**Table 4.** Analysis of heritability of and variance in combining ability for different traits.

| Source of Variation (*df*) | Trait | | | | | | | |
|---|---|---|---|---|---|---|---|---|
| | PH | EY | TY | FNPP | FW | FRS | FH | SSC |
| GCA (9) | 288.33 ** | 2.44 ** | 13.15 ** | 33.53 ** | 3995.85 ** | 68.16 ** | 68.95 ** | 0.39 ** |
| SCA (35) | 36.19 ** | 1.56 * | 1.48 * | 2.55 ** | 553.45 ** | 11.33 ** | 10.22 ** | 0.12 ** |
| Error (88) | 13.12 | 0.89 | 0.95 | 1.03 | 183.72 | 3.68 | 4.025 | 0.06 |
| $2\sigma^2_{GCA}/(2\sigma^2_{GCA} + \sigma^2_{SCA})$ | 0.99 | 0.83 | 0.99 | 0.99 | 0.99 | 0.99 | 0.99 | 0.95 |

Notes: GCA—general combining ability, SCA—specific combining ability, NSH—narrow-sense heritability, BSH—broad-sense heritability, $\sigma^2_{GCA}$ is the variance in GCA effects derived from the mean square of GCA, and $\sigma^2_{SCA}$ is the variance in SCA effects derived from the mean square of SCA. The symbols * and ** indicate significance at 0.05 and 0.01 probabilities, respectively.

When combining ability differs significantly between crosses, heritability can be estimated via the GCA and SCA effect values and their variances. The differences in the values for broad-sense heritability (BSH) and narrow-sense heritability (NSH) between the traits were profound (Figure 2). Among the traits, FNPP showed the highest BSH (74.93%) and NSH (62.65%), and EY showed the lowest BSH (25.13%) and NSH (6.16%). The NSH/BSH ratio of each trait except EY was greater than 52%, indicating that the additive genetic effect was dominant; EY had a low NSH/BSH ratio, indicating that it has a high dominance effect.

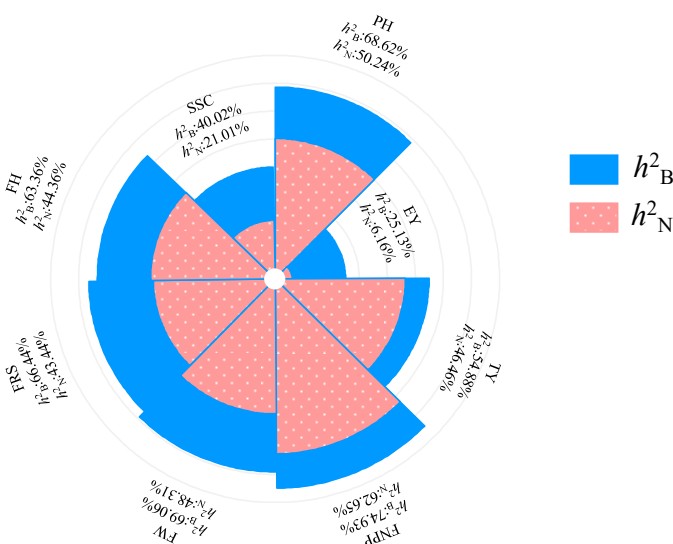

**Figure 2.** Analysis of heritability of different characters. $h^2_N$—narrow heritability estimate, $h^2_B$—broad heritability estimate.

### 3.4. Analysis of the General Combining Ability Effects of Different Traits in Parents

For different parents, GCA differed profoundly for the same trait, while for the same parent, GCA also differed between traits (Figure 3). Comparisons of the GCA values for each parent and trait showed that no parent consistently combined all the traits. The parent 17,969 has a positive effect on all traits except FW and FRS and can be prioritized when selecting parents, especially if the goal is improving yield. Parents 17,955 and 17,927 showed good potential for early maturity. Parent 17,904 was good material for FW. Therefore, when choosing crossing parents, it is necessary to comprehensively consider the GCA effects of different traits and choose parents that show superiority for the desired heterotic traits in accordance with breeding objectives.

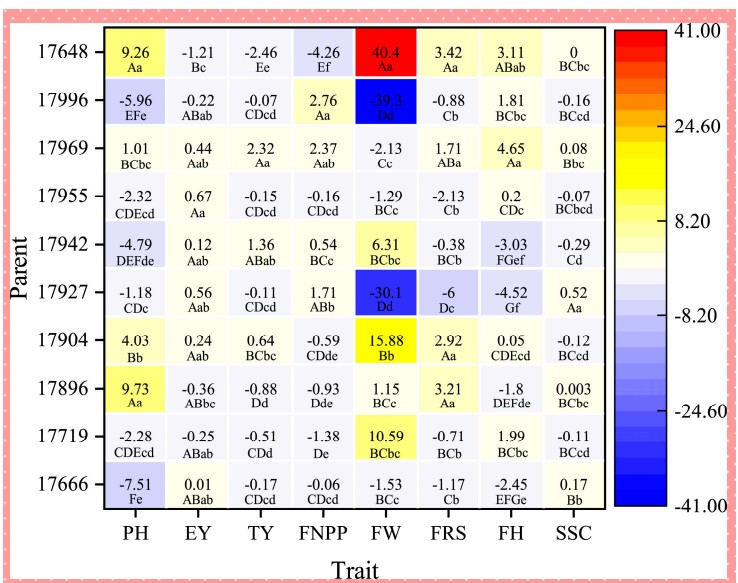

**Figure 3.** General combining ability (GCA) effects in parents for eight main agronomic traits. The vertical axis represents the different parents, and the horizontal axis represents the different traits. The combining ability is coded according to the color key plotted on the right. The blue and red boxes indicate negative and positive combining ability, respectively. For each trait, the parents were compared with each other. In the boxes, different numbers indicate the values of the GCA effects, while lowercase letters and uppercase letters indicate significance at 0.05 and 0.01 probabilities, respectively.

### 3.5. Analysis of the Specific Combining Ability Effects of Crosses for Different Traits

The SCA values of the traits differed profoundly between the crosses (Table 5). For plant height, the 17,896 × 17,996 cross had the highest SCA value, followed by the 17,666 × 17,969 cross, while the 17,896 × 17,969 cross had the highest negative SCA value, followed by the 17,955 × 17,996 cross. For EY, the 17,666 × 17,648 and 17,955 × 17,969 crosses had high SCA values. For TY, the 17,666 × 17,648 cross had the highest SCA value, followed by the 17,927 × 17,969 cross. For FNPP, the 17,996 × 17,648 cross had the highest SCA value, followed by the 17,942 × 17,955 cross. For FW, the 17,666 × 17,648 cross had the highest SCA value, followed by the 17,719 × 17,996 cross. For FRS, the 17,955 × 17,996 cross had the highest negative SCA value, followed by the 17,719 × 17,927 cross. For FH, the 17,969 × 17,648 cross had the highest SCA value, followed by the 17,719 × 17,904 cross. For SSC, the 17,904 × 17,648, 17,719 × 17,927, and 17,666 × 17,969 crosses had high SCA values.

**Table 5.** Specific combining ability (SCA) effects of crosses for the eight main agronomic traits.

| Cross | Trait | | | | | | | |
|---|---|---|---|---|---|---|---|---|
| | PH | EY | TY | FNPP | FW | FRS | FH | SSC |
| 17,666 × 17,719 | −7.13 | 0.14 | 0.38 | 0.57 | −7.01 | −1.79 | −0.90 | 0.09 |
| 17,666 × 17,896 | 5.39 | −0.59 | −0.29 | −1.08 | 6.41 | 2.96 | −0.86 | −0.52 |
| 17,666 × 17,904 | −4.24 | −1.83 | 0.16 | 0.65 | −9.91 | 0.25 | −0.01 | −0.27 |
| 17,666 × 17,927 | −4.76 | 1.16 | −1.04 | −0.38 | −9.04 | −4.50 | −1.08 | −0.003 |
| 17,666 × 17,942 | −3.09 | −0.07 | 0.01 | −0.41 | 4.29 | −3.46 | 0.97 | 0.17 |
| 17,666 × 17,955 | −4.96 | −1.24 | −0.96 | 1.82 | −40.87 | 2.96 | 0.72 | 0.35 |
| 17,666 × 17,969 | 11.11 | −1.61 | −0.74 | −0.38 | −8.66 | −0.21 | −0.72 | 0.37 |
| 17,666 × 17,996 | 1.08 | 1.10 | 0.08 | −0.10 | 1.50 | 2.38 | 1.22 | −0.12 |
| 17,666 × 17,648 | 6.59 | 2.93 | 2.38 | −0.68 | 63.29 | 1.42 | 0.64 | −0.05 |
| 17,719 × 17,896 | −7.83 | 0.31 | 0.30 | −0.43 | 10.37 | 1.17 | 0.19 | −0.25 |
| 17,719 × 17,904 | 5.60 | 0.13 | 0.77 | 1.90 | −19.21 | 0.13 | 6.01 | −0.02 |
| 17,719 × 17,927 | 0.61 | 0.09 | −1.61 | 0.002 | −22.64 | −6.29 | −0.88 | 0.41 |
| 17,719 × 17,942 | 3.49 | −0.67 | −0.51 | −0.56 | 4.93 | −1.92 | −1.41 | −0.82 |
| 17,719 × 17,955 | 6.02 | −0.13 | 1.67 | 1.27 | 1.21 | 2.17 | 0.02 | 0.26 |
| 17,719 × 17,969 | 5.62 | −0.20 | −1.39 | −0.26 | −18.34 | 0.33 | −3.87 | 0.31 |
| 17,719 × 17,996 | −2.88 | 1.48 | 1.08 | −1.65 | 35.20 | 3.92 | 3.40 | 0.09 |
| 17,719 × 17,648 | −3.50 | −1.15 | −0.68 | −0.83 | 15.50 | 2.29 | −2.57 | −0.07 |
| 17,896 × 17,904 | −1.01 | −0.24 | −0.98 | −2.01 | 19.73 | 0.21 | 2.51 | 0.36 |
| 17,896 × 17,927 | 0.74 | 0.17 | 0.06 | 1.82 | −17.26 | −1.54 | 0.73 | 0.09 |
| 17,896 × 17,942 | −1.86 | −0.61 | −0.81 | 0.72 | −22.04 | −1.17 | −1.85 | 0.20 |
| 17,896 × 17,955 | 5.08 | 0.07 | −0.50 | −2.25 | 26.78 | 2.58 | 3.19 | −0.09 |
| 17,896 × 17,969 | −11.59 | 1.83 | 1.01 | 1.02 | −5.92 | −2.25 | −0.22 | −0.004 |
| 17,896 × 17,996 | 12.71 | −1.26 | 1.05 | 1.90 | −9.84 | −1.67 | −0.65 | −0.06 |
| 17,896 × 17,648 | −1.64 | 0.32 | 0.16 | 0.32 | −8.22 | −0.29 | −3.04 | 0.28 |
| 17,904 × 17,927 | 2.90 | −0.47 | 1.14 | 1.42 | −2.12 | 1.75 | −0.20 | −0.29 |
| 17,904 × 17,942 | −1.76 | 1.70 | 0.06 | −0.48 | 9.96 | −0.21 | 0.39 | 0.12 |
| 17,904 × 17,955 | −0.09 | −0.54 | 0.30 | −1.25 | 25.51 | 2.21 | −2.49 | −0.34 |
| 17,904 × 17,969 | −4.09 | 0.94 | −0.70 | −0.05 | −11.47 | −1.96 | −1.07 | 0.22 |
| 17,904 × 17,996 | 6.61 | −0.49 | −0.98 | −0.76 | −4.22 | −1.04 | −3.37 | −0.51 |
| 17,904 × 17,648 | −3.94 | 0.81 | 0.24 | 0.59 | −8.27 | −1.33 | −1.77 | 0.73 |
| 17,927 × 17,942 | 4.59 | −0.04 | 1.89 | 0.35 | 25.42 | 0.38 | 3.15 | 0.29 |
| 17,927 × 17,955 | 2.72 | −1.54 | −1.17 | −1.35 | 1.56 | 0.46 | 2.45 | 0.13 |
| 17,927 × 17,969 | −2.68 | 0.47 | 2.14 | 0.65 | 22.80 | −0.04 | −2.44 | −0.35 |
| 17,927 × 17,996 | −1.05 | 0.47 | −1.94 | −2.86 | 14.90 | 3.54 | 0.21 | 0.19 |
| 17,927 × 17,648 | −3.06 | −0.31 | 0.51 | 0.35 | −13.61 | 6.25 | −1.94 | −0.47 |
| 17,942 × 17,955 | 1.79 | −0.08 | 0.73 | 1.95 | −14.83 | 3.50 | −0.42 | −0.13 |
| 17,942 × 17,969 | 1.66 | 0.34 | 0.99 | 0.09 | 11.76 | 1.33 | −2.50 | 0.23 |
| 17,942 × 17,996 | −5.24 | −0.22 | 0.36 | −0.36 | 12.02 | 2.25 | 1.35 | 0.03 |
| 17,942 × 17,648 | 0.41 | −0.35 | −2.74 | −1.28 | −31.50 | −0.71 | 0.32 | −0.10 |
| 17,955 × 17,969 | 4.19 | 2.60 | 0.41 | 1.65 | −17.30 | 2.75 | −2.31 | 0.03 |
| 17,955 × 17,996 | −9.91 | 1.10 | −0.19 | −0.86 | 7.02 | −10.66 | 0.81 | 0.11 |
| 17,955 × 17,648 | −4.86 | −0.25 | −0.29 | −0.98 | 10.92 | −5.96 | −1.97 | −0.32 |
| 17,969 × 17,996 | −7.77 | −2.29 | −0.80 | −0.26 | −0.67 | 1.50 | −0.10 | −0.27 |
| 17,969 × 17,648 | 3.54 | −2.10 | −0.92 | −2.45 | 27.80 | −1.46 | 13.22 | −0.53 |

### 3.6. Analysis of Midparent Heterosis and Comparison of Crosses for Heterosis in Different Traits

In the $F_1$ generation, the degrees of heterosis in different traits differed between the crosses (Figure 4 and Table 6). The EY and FNPP of most crosses and the TY of all crosses

showed a positive MPH, while the FRS of most crosses showed a negative MPH, also indicating that tomato's yield and early maturity traits had significant heterosis (Table 6). Most crosses had greater values for PH, EY, and FW; lower values for hardness; lower FNPP; and earlier maturity than the control (Table 6).

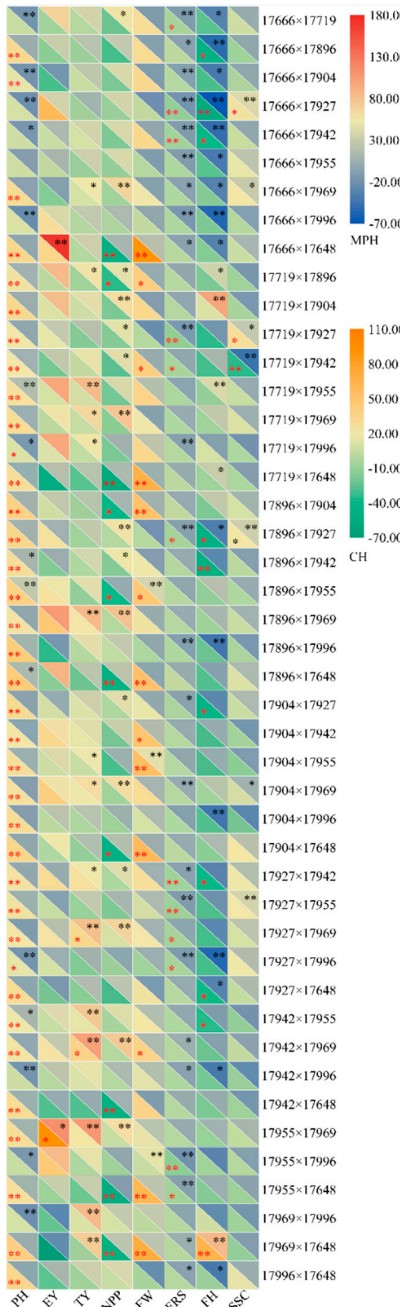

**Figure 4.** Evaluation of heterosis for eight main agronomic traits. The vertical axis represents the different combinations, and the horizontal axis represents the different traits. The degree of heterosis is coded according to the color key plotted on the right. In the boxes, the top right triangle indicates the level of midparent heterosis (MPH); the LSD0.05 and LSD0.01 values of each trait from left to right were 11.21 and 14.85, 110.01 and 145.72, 51.97 and 68.85, 44.57 and 59.04, 29.20 and 38.67, 4.39 and 5.81, 31.01 and 41.07, 25.31 and 33.53. The bottom left triangle indicates the level of comparison heterosis (CH); the $LSD_{0.05}$ and $LSD_{0.01}$ values of each trait from left to right were 12.31 and 16.31, 98.48 and 130.44, 32.36 and 42.86, 33.38 and 44.21, 37.19 and 49.26, 3.86 and 5.12, 34.19 and 45.29, 22.30 and 29.55. The symbols * and ** indicate significance at 0.05 and 0.01 probabilities, respectively.

**Table 6.** Average heterosis of eight main agronomic traits in 45 cross combinations.

| Traits | Mid-Parent Heterosis//% | | | | Comparison Heterosis//% | | | |
|---|---|---|---|---|---|---|---|---|
| | Mean | Range | MPH > 0 | Propotion/% | Mean | Range | CH > 0 | Propotion/% |
| PH | −1.23 | −21.61~27.72 | 20 | 44.44 | 25.06 | 5.40~44.03 | 45 | 100.0 |
| EY | 37.09 | −28.32~173.23 | 38 | 84.44 | 5.97 | −68.46~94.44 | 29 | 64.44 |
| TP | 44.92 | 1.16~108.02 | 45 | 100.0 | 1.49 | −27.59~36.97 | 22 | 48.89 |
| FNPP | 34.72 | −9.00~80.53 | 42 | 93.33 | −17.08 | −55.71~11.42 | 8 | 17.78 |
| FW | 6.13 | −22.83~49.26 | 26 | 57.78 | 28.45 | −6.60~91.34 | 42 | 93.33 |
| FRS | −3.90 | −18.49~5.08 | 15 | 33.33 | −1.70 | −11.59~2.90 | 14 | 31.11 |
| FH | −10.32 | −66.73~100.49 | 17 | 37.78 | −19.62 | −52.47~64.11 | 4 | 8.89 |
| SSC | 2.33 | −44.04~42.08 | 23 | 51.11 | 3.83 | −33.59~28.57 | 26 | 57.78 |

Further analyses of the heterosis of each trait showed that only the 17,896 × 17,955 cross was a tall plant with strong heterosis (MPH = 27.72% and CH = 38.74%) for PH; there were no dwarf plant types with strong heterosis for PH (Figure 4). For EY, the 17,955 × 17,969 cross had strong heterosis (MPH = 114.4% and CH = 94.4%) (Figure 4). For TY, crosses 17,942 × 17,969 and 17,927 × 17,969 showed high MPH and CH, while for FNPP, 17,927 × 17,969 and 17,969 × 17,996 showed both MPH and CH greater than 10% (Figure 4). Therefore, the best performance in terms of yield and FNPP was 17,927 × 17,969, which showed higher yields than did its parents (Figure 5). For FW, the 17,904 × 17,955 cross has strong heterosis (MPH = 47.69% and CH = 55.23%) (Figure 4). For the FRS, the 17,955 × 17,996 cross was highest, followed by crosses 17,666 × 17,927 and 17,719 × 17,927 (Figure 4). For the SSC, crosses 17,666 × 17,927, 17,896 × 17,927, and 17,719 × 17,927 showed good MPH and CH (Figure 4). Overall, the 17,719 × 17,927 cross, followed by the 17,666 × 17,927 cross, outperformed the others on FRS and SSC, while the 17,969 × 17,648 cross has strong heterosis (MPH = 87.11% and CH = 64.11%) for FH.

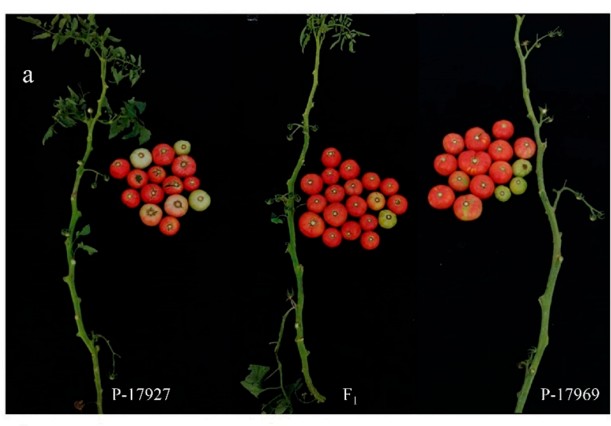

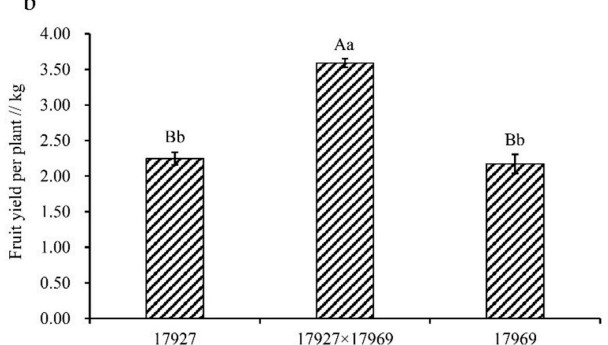

**Figure 5.** Phenotypes of cross 17,927 × 17,969 and its parents. (**a**) Representative plants and corresponding fruit yield from each of the three genotypes. (**b**) Single plant yield data from a randomized block experiment of each of the three genotypes are shown. Single plant yield (kg) is represented on the y-axis. The yields of the three genotypes were compared using a multiple-comparison Tukey test. Uppercase and lowercase letters indicate differences at the 0.01 and 0.05 levels, respectively.

*3.7. Analysis of Midparent Heterosis and Comparison Heterosis of Crosses for Different Traits*

MPH and $F_1$ performance had a highly significantly positive correlation with SCA, indicating that SCA can be used to predict heterosis and hybrid performance (Table 7) and that selecting crosses with high SCA is conducive to obtaining excellent hybrids with development potential. The correlations between GCA and SCA, MPH, and $F_1$ performance were not significant (Table 7), indicating that GCA and SCA are not reliably correlated and that the value of GCA is not sufficient to predict $F_1$ performance or the level of MPH. However, for all the traits, the correlation between the sum of parental GCA values and hybrid performance is basically greater than that between the sum of parental GCA values and the heterosis of the hybrid. The sum of parental GCA is more accurate than SCA for predicting the heterosis of all traits except EY and FW.

**Table 7.** Correlation analysis of combining ability, heterosis and $F_1$ phenotypic value for different traits.

|  | SCA-MPH | SCA-F$_1$ | MPH-F$_1$ | GCA-SCA | GCA-F1 | GCA-MPH | GCAsum-MPH | GCAsum-F$_1$ |
|---|---|---|---|---|---|---|---|---|
| PH | 0.42 ** | 0.57 ** | 0.72 ** | −0.37 | 0.02 | 0.02 | 0.59 ** | 0.82 ** |
| EY | 0.80 ** | 0.84 ** | 0.62 ** | −0.01 | −0.41 | 0.14 | −0.1 | 0.54 ** |
| TP | 0.42 ** | 0.55 ** | 0.76 ** | −0.23 | −0.50 | −0.04 | 0.63 ** | 0.83 ** |
| FNPP | 0.54 ** | 0.48 ** | 0.48 ** | 0.07 | −0.51 | 0.16 | 0.54 ** | 0.65 ** |
| FW | 0.62 ** | 0.59 ** | 0.34 * | −0.48 | −0.51 | −0.48 | −0.03 | 0.81 ** |
| FRS | 0.37 * | 0.63 ** | 0.76 ** | 0.21 | −0.01 | 0.09 | 0.67 ** | 0.78 ** |
| FH | 0.41 ** | 0.60 ** | 0.67 ** | 0.05 | 0.24 | 0.35 | 0.53 ** | 0.80 ** |
| SSC | 0.48 ** | 0.74 ** | 0.92 ** | 0.06 | −0.47 | −0.26 | 0.57 ** | 0.54 ** |

GCAsum, the sum of GCA for two parents. The symbols * and ** indicate significance at 0.05 and 0.01 probabilities, respectively.

## 4. Discussion

Evaluating combining ability and heterosis is the first step in breeding inbred lines to develop commercial hybrids [36]. The genetic bases of heterosis and combining ability remain unclear [37], but this lack of clarity does not negate the importance of heterosis and combining ability in crop breeding. In the utilization of heterosis, the level of combining ability is an important basis for the choice of crossing parents and directly affects the quality of the hybrid [38,39]. Combining ability is measured via two genetic parameters, GCA and SCA, which may be respectively controlled by the additive genetic effects and nonallelic interactions of the parents [40]. The individual traits of different parents have different GCA values, while the SCA values of different parental crosses also differ, suggesting that the additive and the nonadditive genetic effects are fundamentally different. In this study, we found that for all traits, the variances in the GCA of the parents and the SCA of the crosses were all statistically significant and that the $2\sigma^2_{GCA}/2\sigma^2_{GCA}+\sigma^2_{SCA}$ ratio was close to 1. These findings indicate that the traits are affected by the joint action of the additive and nonadditive effects of the genes, and the additive effect of the genes plays a dominant role in the inheritance of the traits [29].

Integrating combining ability, phenotypic performance, and heterosis helps identify crosses with comparatively high levels of heterosis and thus provides valuable insights for crop improvement. For most traits, the GCA effect is strongly correlated with hybrid performance [23], which is inconsistent with the results of this study. The present study showed that the relationship between the GCA effect and hybrid performance was not statistically significant, but the relationship between the sum of parental GCA values and hybrid performance was generally stronger than that between the SCA effect and hybrid performance, except in the cases of the EY and SSC. Therefore, the sum of parental GCA values plays a comparatively important role in predicting hybrid performance in tomato breeding. Heterosis is mainly caused by dominance effects and nonallelic interactions. Therefore, SCA is highly important for heterosis. We found that for eight traits, GCA and SCA did not correlate, which is consistent with the findings of Zhang et al. [17] in rice and Han et al. [18] in soybean. SCA can be used to predict MPH; this conclusion

is also supported by previous studies [41–43]. The results show that $F_1$ heterosis may not directly correlate with parental performance. Heterosis depends on the nature of genetic variation [22]. However, heterosis may depend on the sum of the parents' GCAs and SCA. Through large-scale analysis of combining ability and heterosis of a hybrid maize population, Yu et al. found that the sum of parental GCAs was either negatively correlated or not correlated with heterosis [23]. This finding was inconsistent with the conclusion of the present study, which found that the sum of parental GCA values has a highly significant positive correlation with heterosis for some traits. Combining ability and heterosis relationship is more complex, the establishment of heterotic group might be more help to research heterosis. The utilization of heterosis is the theory basis of the heterotic group and heterosis model, the division of heterotic group and the determination of heterosis pattern by many research results show that for corn, rice, and other crops, there is an obviously improved breeding efficiency, speeding up the progress of the breeding effect [44,45]. The tomato heterotic group division and establishment has been the subject of a few studie; Jin et al. attempted tomato heterosis group division and the establishment of a heterotic group [46]. This will be the future research emphasis.

The early maturity and yield of tomato show significant heterosis [1], and the combination of high yield and early maturity can be obtained through hybridization. This result is consistent with our finding that the EY of most crosses and the TY of all crosses exhibited a positive MPH, while the FRS of most crosses showed a negative MPH. The EY and FRS are important indicators of early maturity in tomato [47]. We found that the variation coefficient of the FRS was low, suggesting that the FRS was relatively stable in all crosses, which is consistent with the result of a previous study showing that a few hybrids may be superior to their parents and their maturity date advances by only 1–5 days [48]. The EY is very important, since it can strongly increase economic efficiency. Therefore, the 17,955 × 17,969 cross, with EY heterosis, would be more valuable to production than the 17,955 × 17,996 cross, which shows heterosis for FRS. In terms of GCA value, 17,969 was superior to 17,996, also indicating that the increase in the yield of the hybrid may result from choosing high-yield parents [49]; this result was also observed in the analysis of heterosis with respect to yield, i.e., 17,942 × 17,969 and 17,927 × 17,969. The EY of tomato negatively correlated with PH and FW but significantly positively correlated with the FNPP, which was consistent with previous results [50]. Therefore, based on PH, FW, and FNPP, we can preliminarily predict the level of the EY. Offspring selection is one of the most important stages in plant breeding. Previous studies have shown that early yield is a trait with low heritability [51–53], which is consistent with our research results, indicating that it was strongly affected by the environment. In pedigree breeding, selection based on EY is not reliable in early generations, but we found that the heritability of the FNPP was high and correlated closely with EY. Therefore, FNPP can be used as an indirect indicator to improve selection efficiency. This result is inconsistent with that of a previous study that found that the EY of tomato is not significantly associated with the environment and that FNPP is ideally used as a selection indicator in late generations [50]; this difference is likely because the two studies used different $F_1$ populations and environments.

The relationship between the components of yield and early maturity is rather complicated. It is difficult to comprehensively balance these factors so that the hybrid has a combination of desirable traits. Nonetheless, the present study has provided an opportunity to select a complex trait through an associated simple trait. To be commercially advantageous, hybrids must be superior to their parents in terms of agronomic traits, especially traits related to yield. Rice and tomato are similar in that most heterotic phenotypes are related to yield [54,55]. In this study, we found that EY had a highly significantly positive correlation with Ty, which is consistent with a previous result [56], indicating that increasing EY helps increase TY. The FNPP had a highly significantly positive correlation with TY and a significantly positive correlation with EY, indicating that heterosis for FNPP is an important indicator for improving yield potential [29]. The contribution of FNPP to TY is greater than that of EY, which is likely because EY affects TY through the FNPP while

also having a direct negative effect on the yield per plant [57]. In some cases, the FNPP is often negatively affected by other traits, such as FW. Therefore, when selecting based on EY, we should pay attention to the FNPP, especially when breeding tomato for high yield. The improvement in FNPP-related traits is an effective way to breed for heterosis for yield. The results and inferences of this study are only based on data from one year and one environment. In order to be more reliable and accurate, the repeated work for many years and in multiple environments will be the focus of the next step.

**5. Conclusions**

The present results showed clear heterosis for precocity and yield in tomato. Both additive and nonadditive genetic effects were involved in the expression of the traits, and the additive genetic effect was dominant in trait inheritance. Although GCA and SCA were not correlated, and the strength of heterosis depends on SCA, GCAsum did predict heterosis for some traits with higher predictive accuracy than did SCA. Compared with heterosis, GCAsum can better predict hybrid performance. Parent 17,969 showed the best overall trait performance, especially in terms of yield. Parents 17,955 and 17,927 showed strong potential for early maturity. The overall characteristics were higher plant height, early maturity, lower early yield, and slightly worse hardness. The best cross for yield per plot and fruit number per plant was 17,927 × 17,969, and the best for fruit ripening stage and soluble solid content was 17,719 × 17,927.

**Author Contributions:** Conceptualization, J.L., H.Y., X.X., H.Z., and T.Z.; methodology, Z.L.; software, Z.L. and Y.S.; validation, Z.L., H.Y. and J.J.; formal analysis, Z.L.; investigation, Z.L., A.R., X.J. and J.J.; resources, J.J.; writing—original draft preparation, Z.L.; writing—review and editing, Z.L.; visualization, Z.L.; supervision, J.J. All authors have read and agreed to the published version of the manuscript.

**Funding:** The study was supported by the Tomato Heterosis Utilization Technology and Strong Dominant Hybrid Seed Creation Project (2016YFD0101703).

**Data Availability Statement:** Germplasm materials is the foundation of breeding, which is valuable for breeders; some special material is the result of years of accumulation. Under normal circumstances, it is not publicly available, but on a non-commercial basis and on an equal exchange basis, it would be available only for research. Materials have their own characteristics in the study, although they were selected from all high-yielding materials. Main features include that the fruit is large, but the number of fruit is less or the fruit is small but the number of fruit is more. In addition, some of the parents containing the *RIN* gene are good late-maturing materials. All materials were provided by the Tomato Research Institute of the College of Horticulture and Landscape Architecture, Northeast Agricultural University, China. Some basic information about the material is shown in Table 1 in the Section 2.

**Conflicts of Interest:** The authors declare no conflict of interest.

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
