# Peer review of "Heterosis and Combining Ability Analysis of Fruit Yield, Early Maturity, and Quality in Tomato"

_agronomy, doi:10.3390/agronomy11040807_

Round 1

Reviewer 1 Report

Dear authors,

most of the results and inferences of your manuscript are sound, but I am in doubt about their novelty, and, in consequence, their interest to a broader audience. I am no tomatoe breeder, but I assume that heterosis for yield, early maturity, and also the trait correlations are already known and well described. To make your manuscript worth being published, you need to:

-highlight better what are the novel findings of your manuscript (if there are)

  • or/and provide a statement on the material: is it or can it be publicly available for other researchers and breeders? Then your work has some merit in describing new germplasm sources.

Otherwise, your work is a sound, but routine evaluation of breeding material, and the detailed descriptions of the performance of different F1-hybrids and combining abilities of number-coded materials are of little interest to other researchers and institutions.

Further general points:

From my point of view there is some misunderstanding on the value of heterosis, e. g. in line 42:

The level of heterosis fundamentally determines the degree of its utilization...

What actually matters is F1 performance, which is: parental mean + heterosis, and can be improved via both factors. Having the highest heterosis does not mean you have the best hybrid, since heterosis tends to be highest in crosses of poor performing parents (in this case, you have strong dominance effects). Further, a positive GCA is not necessarily related to heterosis, it can also be result of a high per se performance. In this regard, you should include the correlation between parental mean and hybrid performance, and between parental per se and respective GCA, in your manuscript.

The relative importance or ratio of GCA vs. SCA does not only depend on the trait, but also on the genetic diversity of the parents. In case of genetically distant parental lines as used in maize breeding (heterotic pools), GCA:SCA ratio is much higher than in crops with less genetic distance among parental pools (e. g. rapeseed, sunflower). You should this discuss how the current situation in tomatoe is, are there heterotic pools like e. g. in maize?

Regarding material & methods:

Your study comprises only one environment, so it is bold to make broader inferences. You need to provide the formula for heritability calculation, from my point of view, having only one environment, no heritability calculation is possible- just a repeatability measure or an estimate of additive vs. dominance effects in diallels like your study.

English language needs to be improved.

Specific points:

line 26:

"Therefore, we should select as many combinations of compatible good characters as possible ac-26 cording to different purposes and needs when breeding new varieties"

this is just common sense among breeders...

line 73: Therefore, when choosing crossing parents, their GCA levels should be determined first 73 to avoid blind selection and should be based on selecting crosses with a high level of SCA 74 so that excellent crosses can be obtained.

you rather mean: selecting crosses with a high level of GCA?

section 3.5:

should be substantially shortened, it is hard to read in its present form. The detailled results regarding SCA should be presented in a table rather in text.

3.6 also here, the detailled results should not be presented in text, hard to read

Altogether, as previously outlined, the detailled material description is only of interest if the material is publicly available.

Line 350: This study showed 350 that the relationship between the GCA effect and hybrid performance was not statistically 351 significant, but relationship between the sum of parental GCA values and hybrid perfor-352 mance was generally stronger than that between the SCA effect and hybrid performance, 353 except in the cases of the early yield and soluble solid content.

I wonder how there can be a relationship between a single GCA effect and hybrid performance. You can only say you have significant GCA effects, i. e. the parental lines differ significantly in their GCA.

Line 357: We found that for eight traits, 357 GCA and SCA did not show any correlation, which is consistent with the findings of 358 Zhang et al. in rice and Han et al. in soybean [15, 18].

For me, this is obvious, since SCA describes deviation from cross performance predicted on GCA (see again line 420)

Line 388: In this study, we found that the heritability of the early 388 yield was relatively low, indicating that it was greatly affected by the environment

How can you make this statement if you had only one environment in this study?

Some cross combinations highlighted in the text seem not be included in the suppl. tables.

best regards

Author Response

Thank you for your help for the improvement to our manuscript entitled “Heterosis and combining ability analysis of fruit yield, early maturity and quality in tomato (agronomy-1158897)”. We are very pleased to learn that our manuscript is acceptable for publication in Agronomy with major revisions. I also want to express our deep thanks to the reviewers of the positive comments and I have revised some based on them.

Now we have studied your comments carefully and have made correction which we hope meet with your approval.

We have modified the manuscript accordingly, and the detailed corrections are listed below point by point:

Reviewer comments

Our responses to the comments

To show our changes, we highlight them with red in the revised manuscript. Please reviewers check them.

Reviewer #1:

1)       I am no tomato breeder, but I assume that heterosis for yield, early maturity, and also the trait correlations are already known and well described.

√ Thank you for reviewer comments. Definitely, but heterosis is complicated. Although ubiquitous, selecting parents based on past performance does not always produce the desired outcome.

We study the yield and ripen of tomatoes based on different materials, and also to meet the market needs for yield and varieties of tomatoes at different ripening stages. Although molecular breeding is on the rise, most breeding is done by conventional cross-breeding. These modern varieties are the result of an intensive plant breeding program in the early 20th century. In order to update varieties and meet the continuous demand of the market, it is necessary to continue to study a lot of different materials and combination resources based on these traits.

2)       Highlight what are the novel findings of your manuscript.

√ Thank you for reviewer comments. First, the novelty is the adoption of materials. Materials have their own characteristics, although they were selected from all high-yielding materials. Main features such as, the fruit is large, but the number of fruit is less or the fruit is small but the number of fruit is more. As well as some of the parents, containing the RIN gene, are good late-maturing materials. The TY-301 was selected, one of the best varieties on the market currently considered by the industry, as the control material.

Here, we conducted the analysis of mid-parent dominance and control dominance, and screened a parent with good comprehensive traits, 17969, a high-yielding candidate combination 17927×17969, and a precocious and good taste candidate combination 17666×17927.

3)       Provide a statement on the material: is it or can it be publicly available for other researchers and breeders?

√ Germplasm materials is the foundation of breeding, which is valuable for breeders, especially some special material, is the result of years of accumulation. Under normal circumstances is not publicly, but on a non-commercial basis and on an equal exchange basis, it would be available only for research.

4)       What actually matters is F1 performance, which is: parental mean + heterosis, and can be improved via both factors. Having the highest heterosis does not mean you have the best hybrid, since heterosis tends to be highest in crosses of poor performing parents (in this case, you have strong dominance effects)?

√ Thank you very much for your kind advice. For this condition, We selected an excellent variety TY-301 as the control, carried on the control heterosis analysis, in order to improve and guarantee the commodity value of the selected combination of advantages.

5)       The relative importance or ratio of GCA vs. SCA does not only depend on the trait, but also on the genetic diversity of the parents. In case of genetically distant parental lines as used in maize breeding (heterot5ic pools), GCA:SCA ratio is much higher than in crops with less genetic distance among parental pools (e. g. rapeseed, sunflower). You should this discuss how the current situation in tomatoes is, are there heterotic pools like e. g. in maize?

√ Thank you very much for your kind advice. There is a small part in the discussion about tomato heterotic group. Combining ability and heterosis relationship is more complex, the establishment of heterosis group might be more help to research heterosis. The tomato heterosis group division and heterosis group of few studies. Utilization of heterosis is the theory basis of heterosis group and heterosis model, the division of heterosis group and the determination of heterosis pattern by many research results show that for corn, rice and other crops breeding has obvious improve breeding efficiency, speed up the progress of the breeding effect. This may be a research trend in the tomato.

6)       Your study comprises only one environment, so it is bold to make broader inferences. You need to provide the formula for heritability calculation, from my point of view, having only one environment, no heritability calculation is possible- just a repeatability measure or an estimate of additive vs. dominance effects in diallels like your study?

√ Thank you very much for your responsible comments. The following formula is usually used to calculate heritability: H2=VG (Genetic variance)/ VP (Phenotypic variance); h2=VA (Additive variance)/ VP (Phenotypic variance).

Here, broad heritability and narrow heritability are estimated by Statistical analysis program of genetic mating design under professional statistics module in the statistical software DPS (Data Processing System), which is a data processing and analysis software independently developed by Zhejiang University in China. More than 600 functions are found in DPS, including experimental design, statistical analysis and data mining [1]. The combining ability effect of the test materials was estimated according to the fixed model (model Ⅰ), and the various variance components and related genetic parameters were estimated according to the random model (model Ⅱ) [2-5]. Yes, we are also continuing research, the selection of varieties is a long process. For its genetic stability and reliability, the test is currently being repeated in multi-regional environments and many years.

References

1.      Tang, Q.Y., Zhang, C.X. Data Processing System (DPS) software with experimental design, statistical analysis and data mining developed for use in entomological research. Insect Science, 2012, 00, 1–7. DOI 10.1111/j.1744-7917.2012.01519.x

2.      Qi, S.W.; Sheng, X.B. Analysis on Combining Ability and Heritability of Major Agronomic Characters in Two-line Indica Hybrid rice. Hybrid Rice, 2000, 15(3), 38-40.

DOI: 10.16267/j.cnki.1005-3956.2000.03.020

3.      Mo, H.D. The Analysis of Combining Ability in P × q Mating Pattern. Jiangsu Agricultural Research, 1982, (03), 51-57.

DOI: 10.16872/j.cnki.1671-4652.1982.03.008

4.      Zhou, K.D.; Li, H.Y.; Li, R.D.; Luo, G.J. A Preliminary Study on The Heritability and Combining ability of The Major Characteristics in Hybrid Rice. Acta Agronomica Sinica, 1982, (03), 145-152.

5.      Xu, J.F. et al., editor. Quantitative genetics and rice breeding. Anhui science & technology press, Hefei, 1990, pp.116-172.

7)       English language needs to be improved.

√ Thank you very much for your responsible comments. We have checked the English expression and improve it.

8)       line 73: Therefore, when choosing crossing parents, their GCA levels should be determined first to avoid blind selection and should be based on selecting crosses with a high level of SCA so that excellent crosses can be obtained.

you rather mean: selecting crosses with a high level of GCA?

√ Thank you for the professional comments of reviewer. Yes, select parents with high general combining ability. High general combining ability indicates that the genes of quantitative traits have a strong cumulative effect [1]. In general, high special combining ability is selected on the basis of high general combining ability, and then it is easier to obtain the highest combination [2]. Such as, for yield traits, those with high general combining ability were selected, while for early maturity traits, those with high negative general combining ability were selected.

References

1.      Fang, Z.Y.; Editor in chief. Vegetable Breeding in China. China Agriculture Publishing House, Beijing, China, 2017; pp.1009

2.      Li, J.F. Editor in chief. Chinese Tomato Breeding. China Agriculture Publishing House, Beijing, China, 2011; pp.141

9)       Section 3.5: should be substantially shortened, it is hard to read in its present form. The detailled results regarding SCA should be presented in a table rather in text.

3.6 also here, the detailled results should not be presented in text, hard to read

√ Thank you for the professional comments of reviewer. This section was modified according to your kind suggestion. Corresponding table has been added.

10)    Line 350: This study showed that the relationship between the GCA effect and hybrid performance was not statistically significant, but relationship between the sum of parental GCA values and hybrid performance was generally stronger than that between the SCA effect and hybrid performance, except in the cases of the early yield and soluble solid content.

I wonder how there can be a relationship between a single GCA effect and hybrid performance. You can only say you have significant GCA effects, i. e. the parental lines differ significantly in their GCA.

√ Thank you for the professional comments of reviewer. Many studies have shown that general combining ability is related to hybrid performance. Such as, the GCA effect of most traits strongly correlated with the performance of the hybrid, suggesting that parental GCA may be a good predictor of the performance of the hybrid [1]. Fischer et al. found in maize that the performance of maize hybrids mainly determined by the GCA effect [2]. Choudhary et al. also observed that better per-forming crosses had at least one of the cultivars with high GCA effect [3]. Therefore, a correlation analysis of general combining ability and hybrid performance was carried out in tomato.

Reference:

1.      Yu, K.C.; Wang, H.; Liu, X.G.; Xu, C.; Li, Z.W.; Xu, X.J.; Liu, J.C.; Wang, Z.H.; Xu, Y.B. Large-Scale Analysis of Combining Ability and Heterosis for Development of Hybrid Maize Breeding Strategies Using Diverse Germplasm Resources. Frontiers in plant science, 2020, 11, 660. (DOI: 10.3389/fpls.2020.00660)

2.      Fischer, S.; Möhring, J.; Schön, C.C.; Piepho, H.P.; Klein, D.; Schipprack, W.; Utz, H.F.; Melchinger, A.E.; Reif, J.C. Trends in genetic variance components during 30 years of hybrid maize breeding at the University of Hohenheim. Plant Breeding, 2009, 127, 446-451. (DOI: 10.1111/j.1439-0523.2007.01475.x)

3.      Choudhary, A.K.; Chaudhary, L.B.; Sharma, K.C. Combining ability estimate of early generation inbred lines derived from two maize populations. Indian Journal of Genetics and Plant Breeding, 2000, 60(1), 55–61.

11)    Line 388: In this study, we found that the heritability of the early yield was relatively low, indicating that it was greatly affected by the environment

How can you make this statement if you had only one environment in this study?

√ Thank you for the professional comments of reviewer. The various variance components and related genetic parameters were estimated according to the random model (model Ⅱ) [1-4]. The results indicate that early yield is a trait with low heritability, which is consistent with other early studies [5-7]. This also shows the reliability of our results. Repeated studies under multiple environmental conditions can indeed increase the validity and authenticity of the result data, and this study is also doing this work.

References

1.      Qi, S.W.; Sheng, X.B. Analysis on Combining Ability and Heritability of Major Agronomic Characters in Two-line Indica Hybrid rice. Hybrid Rice, 2000, 15(3), 38-40. (DOI: 10.16267/j.cnki.1005-3956.2000.03.020)

2.      Mo, H.D. The Analysis of Combining Ability in P × q Mating Pattern. Jiangsu Agricultural Research, 1982, (03), 51-57. (DOI: 10.16872/j.cnki.1671-4652.1982.03.008)

3.      Zhou, K.D.; Li, H.Y.; Li, R.D.; Luo, G.J. A Preliminary Study on The Heritability and Combining Ability of The Major Characteristics in Hybrid Rice. Acta Agronomica Sinica, 1982, (03), 145-152.

4.      Xu, J.F. et al., editor. Quantitative genetics and rice breeding. Anhui science & technology press, Hefei, 1990, pp.116-172.

5.      Hu, Q.D.; Zhang, H.Q.; Li, C.L. et al. A Preliminary Study on the Genetic Parameters of Tomato's Main Early Maturity Traits and Their Application in Breeding. China Vegetables, 1990, (01), 4-7.

6.      Cui, H.W.; Deng, J.J. The Combining Ability of Several Quantitative Characters of Cuvumber Celfbred Parent Lines and Their Inheritance Analysis. Journal of Northwest A & F University (Natural Science Edition), 1987, (30), 63-71.

7.      Wang, D.Y.; Kong, Z.H.; Wang, M.; Zhou, X.M. Study on Genetic Parameters of Main Early Maturity Characters in Pepper. Shaanxi Journal of Agricultural Sciences, 1993, (05), 17-19.

We greatly appreciate your help in improving this paper and trust that the revised manuscript is now suitable for the journal's desired standard. If there is any shortage, please inform us! We will try our best to revise this manuscript. Thank you very much for your attention and kind advice.

With best regards,

Jingfu Li

Reviewer 2 Report

none

Author Response

Response to Reviewer Comments

Thank you for your help for the improvement to our manuscript entitled “Heterosis and combining ability analysis of fruit yield, early maturity and quality in tomato (agronomy-1158897)”. 

We greatly appreciate your help in improving this paper and trust that the revised manuscript is now suitable for the journal's desired standard. If there is any shortage, please inform us! We will try our best to revise this manuscript. Thank you very much for your attention and kind advice.

With best regards,

Jingfu Li

Reviewer 3 Report

This paper evaluates 45 crosses of tomato lines in order to find new variety candidates utilizing the advantages of heterosis. For this, basic and easily measurable traits were selected. The demonstration of the results is fine, however, the materials and methods part requires further clarifications. The method applied seems to be more powerful than the dataset used for the demonstration.

General comments

The criteria of selecting the eight descriptive traits is missing from the MS. These traits are very important, but it is questionable for me whether these are sufficient for selecting the best crosses. The article does not concern issues related to plant protection issues or abiotic disorders. The authors say that major agronomic traits were selected and measured, however, the ratio of marketable fruits is also a very important data. Why weren’t these topics evaluated?

Detailed comments:

line 188: How do you mean ’under a shed’? Please clarify.

line 122: after the harvest of the early yield

line 123: abbreviations should be defined at the first use, please correct it.

Line 128: averaged height data means that you will have one value per plot? Was that for reducing the standard deviation of the dataset?

Line 129: How did you select the fruits which were evaluated? Were there any exclusion criteria for fruits, like symptoms of infection, cracking, scars? If yes, please add this to the MS.

Line 133: How do you define full fruit period? Please add.

Line 138: Where did you measure fruit firmness? Top, bottom, sides/largest circumference, center line? Please add.

Line 143: I suspect, this is an Atago PAL-1 instrument. Please add. How do you mean well-mixed juice? How were the homogenates prepared? Please add.

Line 181: FRS refers to first ripening stage, not fruit ripening stage. The same mistake in line 183.

Fig 1: line 189: hybrids. Abbreviations are not defined in the captions. N is not given here, which is an important factor of the reliability of correlation analysis.

Fig 3: indicating significant differences – here it is not defined, where the significant difference is. For example, upper case between lines, lower case between traits, I suspect.

Line 361: Sentence starting with GCA has no … please reword or shorten.

Line 366: study where the parental GCA

Line 376: study where a few hybrids

Author Response

Response to Reviewer Comments

Thank you for your help for the improvement to our manuscript entitled “Heterosis and combining ability analysis of fruit yield, early maturity and quality in tomato (agronomy-1158897)”. We are very pleased to learn that our manuscript is acceptable for publication in Agronomy with major revisions. I also want to express our deep thanks to the reviewers of the positive comments and I have revised some based on them.

Now we have studied your comments carefully and have made correction which we hope meet with your approval.

We have modified the manuscript accordingly, and the detailed corrections are listed below point by point:

To show our changes, we highlight them with red in the revised manuscript. Please reviewers check them.

1. The criteria of selecting the eight descriptive traits is missing from the MS. These traits are very important, but it is questionable for me whether these are sufficient for selecting the best crosses. The article does not concern issues related to plant protection issues or abiotic disorders. The authors say that major agronomic traits were selected and measured, however, the ratio of marketable fruits is also a very important data. Why weren’t these topics evaluated?

Our responses to the comments:

Thank you for the professional comments of reviewer. It takes many years of effort to breed a variety suitable for the market with high yield and good quality. This study is only the first step in the selection work. The excellent varieties currently on the market have been bred from many candidate combinations through conventional crossbreeding and observation over many years. The TY-301 is a recognized and official variety that is currently selling well on the market. As a comparison, it is hoped that a variety comparable to or superior to this variety will be developed. At present, it is only in the field observation stage, mainly for yield and ripening. For some traits, there are some combinations superior to the control in this study, which can accumulate a certain resource base for the next step of cross breeding. However, further observation on the stability of traits is still needed. As you said, this is indeed an important trait, and we will work on this aspect in the later stage.

2. Line 188: How do you mean ’under a shed’? Please clarify.

Our responses to the comments:

Thank you for the professional comments of reviewer. All materials are planted in plastic greenhouses and their normal field management is under the greenhouses.

3. Line 122: after the harvest of the early yield

Our responses to the comments:

Thank you for the professional comments of reviewer. It was modified. It should be the early yield after harvest. We have changed “after the harvest of the early yield” to “the early yield after the harvest”

4. Line 123: abbreviations should be defined at the first use, please correct it.

Our responses to the comments:

Thank you for the comments of reviewer. We have corrected it.

5.  Line 128: averaged height data means that you will have one value per plot? Was that for reducing the standard deviation of the dataset?

Our responses to the comments:

Thank you for the comments of reviewer. Yes, we measured five plants in each plot, repeated three times.

6. Line 129: How did you select the fruits which were evaluated? Were there any exclusion criteria for fruits, like symptoms of infection, cracking, scars? If yes, please add this to the MS.

Our responses to the comments:

Thank you for the comments of reviewer. This is the total number of fruits from the first, second, third, and fourth inflorescences at maturity. The selected fruits were all smooth and complete or had slight mechanical damage. Deformed fruits infected with disease, rotten fruits and fruits with substantial cracking were not included in the statistical analysis. Due to good management, leaf thinning and ripe fruit harvesting were achieved in time, with less rotten fruit and cracked fruit.

Original: This is the total number of fruits from the first, second, third, and fourth inflorescences at maturity.

7. Line 133: How do you define full fruit period? Please add.

Our responses to the comments:

Thank you for the comments of reviewer. This is the total yield after harvesting at 4-day intervals prior to the full fruit period, the period when the yield in the plot was greatest.

Original: This is the yield before the full fruit period.

8.  Line 138: Where did you measure fruit firmness? Top, bottom, sides/largest circumference, center line? Please add.

Our responses to the comments:

Thank you for the comments of reviewer. A peel sample of approximately 1 cm2 was removed with a blade at a 120° angle between the shoulder of each fruit. A probe of 1 cm2 was selected, and a hand-held durometer (HANAPI, MODEL GY-4) was used to measure according to the manufacturer’s instructions. Five complete fruits were randomly selected from each plot, and the average value was considered the hardness value of the fruit in the plot; this process was repeated 3 times.

Original: Each fruit was measured with a hand-held hardness meter (HANAPI, MODEL: GY-4) three times at an interval of 120°, and the average value of the three measurements was taken. From each plot, five fruits were randomly sampled for the measurement.

9. Line 143: I suspect, this is an Atago PAL-1 instrument. Please add. How do you mean well-mixed juice? How were the homogenates prepared? Please add.

Our responses to the comments:

Thank you for the comments of reviewer. At the full fruit period, five fruits randomly chosen for each plot. Cut each fruit crosswise, squeeze out the right amount of juice from the crosswise by hand and put it into a clean and waterless container. Take about the same amount of liquid from five fruits and mix well.

10. Line 181: FRS refers to first ripening stage, not fruit ripening stage. The same mistake in line 183.

Our responses to the comments:

Thank you for the comments of reviewer. We have corrected them. We have changed “fruit” to “first”.

11.  Fig 1: line 189: hybrids. Abbreviations are not defined in the captions. N is not given here, which is an important factor of the reliability of correlation analysis.

Our responses to the comments:

Thank you for the comments of reviewer. According to the suggestion of reviewer, Now we have added.

12.  Fig 3: indicating significant differences – here it is not defined, where the significant difference is. For example, upper case between lines, lower case between traits, I suspect.

Our responses to the comments:

Thank you for the comments of reviewer. For each trait, the parents were compared with each other. In the boxes different number indicate general combining ability (GCA) effects value, while lower case letters and upper case letters indicate significance at 0.05 and 0.01 probability levels, respectively.

13. Line 361: Sentence starting with GCA has no … please reword or shorten.

Our responses to the comments:

Thank you for the comments of reviewer. We have corrected it.

14.  Line 366: study where the parental GCA and a few hybrids.

Our responses to the comments:

Thank you for the comments of reviewer. May be the meaning of the sentence is not clear, which has caused misunderstanding. Now I rearrange this paragraph as follows, hoping it can meet your needs.

Through large-scale analysis of combining ability and heterosis of hybrid maize population, Yu et al. found that the sum of parental GCAs was either negatively or not correlated with heterosis. This is is inconsistent with the conclusion of this study that the sum of parental GCA values has a highly significant positive correlation with heterosis on some traits. This may be because the two studies were conducted on different crop populations and combining ability had different effects on different crops or traits.

We greatly appreciate your help in improving this paper and trust that the revised manuscript is now suitable for the journal's desired standard. If there is any shortage, please inform us! We will try our best to revise this manuscript. Thank you very much for your attention and kind advice.

With best regards,

Jingfu Li

Round 2

Reviewer 1 Report

Dear authors,

thank you for your replies, I am willing to accept your paper in the present form after a few minor but still important complementations:

Please include a statement on the plant material (background, possibility of sharing with other institutions), similarly as you did in the authors reply, in the manuscript.

Please write "heterotic group" instead "heterosis group"

The shortcomings of your paper, i. e. having only one environment, and hence the possibility of only limited inferences, should be mentioned in conclusions section.

best regards

Author Response

Response to Reviewer Comments

Thank you for your help for the improvement to our manuscript entitled “Heterosis and combining ability analysis of fruit yield, early maturity and quality in tomato (agronomy-1158897)”. We are very pleased to learn that our manuscript is acceptable for publication in Agronomy with minor revisions. I also want to express our deep thanks to the reviewers of the positive comments and I have revised some based on them.

Now we have studied your comments carefully and have made correction which we hope meet with your approval.

We have modified the manuscript accordingly, and the detailed corrections are listed below point by point:

To show our changes, we highlight them with red in the revised manuscript. Please reviewers check them.

1、Please include a statement on the plant material (background, possibility of sharing with other institutions), similarly as you did in the authors reply, in the manuscript.

Thank you for reviewer comments. Statement on the plant material has been added in the manuscript.

Germplasm materials is the foundation of breeding, which is valuable for breeders, especially some special material, is the result of years of accumulation. Under normal circumstances is not publicly, but on a non-commercial basis and on an equal exchange basis, it would be available only for research. Materials have their own characteristics in the study, although they were selected from all high-yielding materials. Main features such as, the fruit is large, but the number of fruit is less or the fruit is small but the number of fruit is more. As well as some of the parents, containing the RIN gene, are good late-maturing materials. All materials were provided by the Tomato Research Institute of the College of Horticulture and Landscape Architecture, Northeast Agricultural University, China. Some basic information about the material is shown in Table 1 in the material method section.

2、Please write "heterotic group" instead "heterosis group"

Thank you for reviewer comments. We have changed "heterotic group" instead "heterosis group"

3、The shortcomings of your paper, i. e. having only one environment, and hence the possibility of only limited inferences, should be mentioned in conclusions section.

Thank you very much for your responsible comments. The results and inferences of this study are only based on data from one year and one environment. In order to be more reliable and accurate, the repeated work for many years and in multiple environments will be the focus of the next step. This has been added to the discussion section.

We greatly appreciate your help in improving this paper and trust that the revised manuscript is now suitable for the journal's desired standard. If there is any shortage, please inform us! We will try our best to revise this manuscript. Thank you very much for your attention and kind advice.

With best regards,

Jingfu Li
